# Association of *Helicobacter pylori* as an Extragastric Reservoir in the Oral Cavity with Oral Diseases in Patients with and Without Gastritis—A Systematic Review

**DOI:** 10.3390/microorganisms13081955

**Published:** 2025-08-21

**Authors:** Eber Cuba, María C. Sánchez, María J. Ciudad, Luis Collado

**Affiliations:** 1Department of Medicine, Faculty of Medicine, Complutense University, 28040 Madrid, Spain; ebercuba@ucm.es (E.C.); mariasan@ucm.es (M.C.S.); mjciudad@ucm.es (M.J.C.); 2GINTRAMIS Research Group (Translational Research Group on Microbiota and Health), Faculty of Medicine, Complutense University, 28040 Madrid, Spain

**Keywords:** *Helicobacter pylori*, oral cavity, extragastric reservoir, oral diseases, gastritis

## Abstract

This systematic review aimed to evaluate the association between *Helicobacter pylori* in the oral cavity as an extragastric reservoir and oral diseases in patients with or without gastritis. Following the PRISMA guidelines, a comprehensive search was conducted on the PubMed, Scopus, Cochrane Central, and Embase databases (2010–2025) using MeSH terms and keywords related to *H. pylori*, the oral cavity, and oral diseases. Inclusion criteria included observational studies, clinical trials, and case–control studies. Data extraction and quality assessment were performed using the Newcastle–Ottawa Scale (NOS). Of the 298 records initially identified, 22 studies met the inclusion criteria. The presence of *H. pylori* in the oral cavity (plaque, saliva) was variably associated with gastritis, periodontitis, dental caries, and halitosis. Detection rates varied widely (0–100%), influenced by methodological differences (PCR, culture, antigen tests). Some studies reported an improvement in oral health after eradication therapy, while others found no significant association. The oral cavity may serve as a reservoir for *H. pylori*, with implications for oral and systemic health. Standardized diagnostic methods and integrated treatment approaches (combining gastric eradication and oral hygiene) are needed to clarify their role and optimize clinical outcomes. Further research is warranted to establish causal relationships and therapeutic strategies.

## 1. Introduction

*Helicobacter pylori* is a Gram-negative bacterium that infects the stomach and is associated with various gastrointestinal diseases, including gastritis, peptic ulcers, and gastric cancer [1,2]. Although the main colonization of *H. pylori* occurs in the stomach, its presence has been detected in other regions of the body, including the oral cavity. The oral cavity, as a gateway to the digestive tract, is home to a diverse microbiome that plays a crucial role in oral and systemic health. The presence of *H. pylori* in the oral cavity raises questions about its role as an extragastric reservoir and its possible involvement in the development of oral diseases [3,4,5,6].

The prevalence of *H. pylori* in the oral cavity varies considerably between studies, which can be attributed to differences in methodology and the populations studied and the presence of risk factors such as poor oral hygiene and low socioeconomic status [3,4,5].

A great challenge in the study of *H. pylori* in the oral cavity is its precise detection and the differentiation of potentially pathogenic strains from those that are part of the normal oral microbiome. The bacteria may be present in low concentrations and may be difficult to grow from oral samples [4,7]. In addition, the presence of genetically similar bacteria in the oral cavity can lead to false-positive results [4].

The relationship between *H. pylori* as an extragastric reservoir and oral diseases is complex and not yet fully elucidated. Several studies have investigated the associations between *H. pylori* infection and specific oral diseases, such as periodontitis [3,5], halitosis [6,8], caries [9,10,11], and oral lichen planus [12], but the results have been inconsistent.

In previous studies, the accumulated evidence suggests that the oral cavity may act as an extragastric reservoir for *H. pylori*, complicating eradication efforts and increasing the potential implications for oral health [4,9,10,11]. The bacterium has been detected at several oral sites, including dental plaque, saliva, and dental pulp, suggesting a possible route of transmission and a complex interaction with the oral microbiome [5,9,11].

The genetic makeup of *H. pylori* can also influence its ability to colonize the oral cavity. An important virulence factor, associated with pathogenicity of *H. pylori*, has been detected in oral strains and has been linked to an increased adhesion capacity and inflammation. Other virulence genes, such as cow, have also been investigated in oral strains, but their role in oral pathogenesis is not yet fully understood [7].

The presence of *H. pylori* in the oral cavity has been associated with various oral diseases, such as periodontitis, halitosis, and dental caries [3,4,9,10]. Periodontitis, a chronic inflammatory disease that affects the supporting tissues of the teeth, has been linked to the presence of *H. pylori* both in the stomach and in the oral cavity. It has been hypothesized that *H. pylori* may contribute to periodontitis by inducing a local inflammatory response and upsetting the balance of the oral microbiome [3,9]. Tooth decay, although mainly associated with other factors, has also been linked to the presence of *H. pylori* in the oral cavity, suggesting a possible role of the bacteria in the demineralization of tooth enamel [4,6].

Other research has explored the relationship between *H. pylori* and other oral diseases, such as oral leucoplakia and oral lichen planus, but evidence is still limited [12]. It has been shown that infection by *H. pylori* alters the composition of the oral microbiome, leading to changes in the abundance of specific bacteria [1,5,9,11]. These changes in the oral microbiome may contribute to the development of oral diseases by creating a more favorable environment for the growth of pathogenic bacteria or by affecting the local immune response [1,5].

Research has also been carried out regarding the impacts of *H. pylori* eradication on oral health. Some studies have reported an improvement in oral health parameters, such as reduced periodontal probing depths and improved gingival bleeding [3,5]. However, other studies have not found a significant association between the eradication of *H. pylori* and an improvement in oral health [4].

On the other hand, gastroesophageal reflux disease (GERD) could play a relevant role in the colonization of the oral cavity by *H. pylori*. The reflux of gastric contents into the oral cavity can transport viable bacteria from the stomach, facilitating their implantation in the oral mucosa, dental plaque, or periodontal pockets. This phenomenon could explain the presence of this bacterium in individuals without direct oral–oral contact, as well as its finding in patients with untreated or poorly controlled GERD [13].

In addition, when the environment is altered by repeated exposure to gastric acid, this could modify the oral microbiota, reducing bacterial competition and favoring the persistence of *H. pylori* [14]. Some studies even suggest that the prevalence of *H. pylori* via the oral transmission route is higher in patients with chronic reflux symptoms, which reinforces the hypothesis of a gastro–oral transmission route. Thus, GERD could not only constitute a risk factor for oral colonization by *H. pylori* but also a key pathophysiological mechanism connecting gastric infection and extragastric reservoirs [15].

The systematic review presented here is based on the need to clarify the association between the presence of *H. pylori* in the oral cavity and the development of oral diseases, as well as the ability of the oral cavity to act as a potential extragastric reservoir of *H. pylori* in both non-gastritis and gastritis patients. Although there are multiple studies that address the detection of the bacterium in various locations, to date, there has not been a consensus regarding a full understanding of the role of the oral cavity in the persistence and spread of infection. Determining this association is critical to optimize therapeutic strategies, ensuring that treatment with *H. pylori* addresses all possible reservoirs, reducing the rate of reinfection and improving clinical outcomes. This is also necessary to prevent oral complications, identifying patients at risk of developing diseases associated with bacterial presence and establishing specific preventive measures, as well as to expand knowledge. A scientific approach has been developed by consulting the existing literature and integrating data to allow a more global view of the relationship between the oral colonization of *H. pylori* and alterations in oral health, which could open up new lines of research in the field of gastroenterology and dentistry.

For this reason, the objective of this review was to carry out a critical synthesis of the scientific literature published in the last 15 years to determine the prevalence of the oral cavity as an extragastric reservoir of *H. Pylori* and its associations with different diseases, such as caries, periodontitis, and supra- and subgingival biofilms.

## 2. Materials and Methods

### 2.1. Development and Registration of the Protocol

We conducted a comprehensive systematic review of the published literature according to the Preferred Reporting Items for Systematic Reviews and Meta-Analyses (PRISMA) 2020 guidelines (Appendix A) [16]. The methodological protocol was registered in the International Prospective Registry of Systematic Reviews (PROPERO) with the registration number CRD420251089644, ensuring transparency and adherence to the standards established for systematic reviews in answering the research question on the association between *H. pylori* in the oral cavity and oral diseases in patients with or without gastric infection. The PECO criteria were defined, establishing the population, exposure, comparators, and results of interest, and a comprehensive search strategy was designed for electronic databases using Medical Subject Headings (MeSH) terms and keywords. Inclusion and exclusion criteria were applied to select studies published between 2010 and 2025, and tools such as the Newcastle–Ottawa Scale (NOS) were used to evaluate methodological quality.

### 2.2. Research Question

Is there an association between the presence of *H. pylori* in the oral cavity and the incidence of oral diseases (such as dental caries, periodontitis, halitosis, and peri-implantitis) in patients with or without gastric infection?

### 2.3. PECO Criteria

Population (P): Patients of any age, sex, or ethnicity, with or without a diagnosis of gastritis or gastric infection by *H. pylori*. Participants screened for *H. pylori* in the oral cavity, either in saliva, dental plaque, biofilm, or oral tissue, are included.

Exposure (E): The presence of *H. pylori* in the oral cavity, detected by specific diagnostic methods such as polymerase chain reaction (PCR), microbiological cultures, antigen tests, or serological tests.

Comparator (C): Patients without the presence of *H. pylori* in the oral cavity and/or without a diagnosis of gastritis or active gastric infection.

Results (O): Association between the presence of *H. pylori* in the oral cavity and the development of oral diseases, including dental caries, periodontitis, peri-implantitis, halitosis, and alterations of the oral microbiome. Differences in the incidence of oral diseases in patients with and without gastric infection will also be considered.

### 2.4. Search Strategy

The search strategy was systematically and comprehensively designed to identify relevant studies published between 2010 and 2025. We searched key electronic databases such as PubMed, Scopus, Embase, and Cochrane Central using a combination of MeSH terms such as *H. pylori*, caries, saliva, biofilm, periodontitis, peri-implantitis, and oral cavity, as well as *H. pylori*-related keywords such as oral cavity, oral diseases (dental caries, periodontitis, peri-implantitis), and oral microbiome. Search terms included variants such as “oral *Helicobacter pylori*”, “periodontal disease AND *H. pylori*”, and “biofilm AND *H. pylori*”, among others. The strategy was adapted to the specificities of each database to maximize the retrieval of relevant studies. In addition, we reviewed the reference lists of selected articles and hand-searched peer-reviewed journals to identify additional studies that may have been missed in the electronic searches. Finally, the search formula obtained was as follows:

“*Helicobacter* AND *pylori* AND caries OR saliva OR biofilm AND oral periodontitis OR peri-implantitis OR oral cavity”.

Filters were applied to limit the results to human studies, published in English or Spanish, that met predefined inclusion criteria. The search results were exported to a reference manager (Zotero) to facilitate the removal of duplicates and the organization of the identified studies. Subsequently, screening was carried out in two stages: first, titles and abstracts were examined to exclude clearly irrelevant studies, and then the full texts of the remaining studies were assessed to confirm their eligibility according to the PECO criteria. This methodology ensured the rigorous and reproducible identification of the available literature on the subject.

### 2.5. Selection Criteria

-Inclusion criteria

For this systematic review, we included studies published between 2010 and 2025, where the study participants were of any age, sex, or ethnicity. Studies that adhered to the PECO framework in examining the relationships between dental caries, periodontitis, peri-implantitis, saliva, and biofilms and *H. pylori* in patients with and without gastric infection, including observational case–control, cohort, cross-sectional, and clinical trial articles, were included for analysis. By including studies that met these criteria, the systematic review aimed to obtain a comprehensive understanding of the current literature on the association between *H. pylori* in the oral cavity and oral diseases in patients with and without gastric infection. We also considered the type of diagnostic method used to identify *H. pylori*. Studies were published in English or Spanish, approved by ethics committees, and provided relevant data.

-Exclusion Criteria

Studies published before 2010 were excluded, as well as articles consisting solely of titles, conference abstracts, editorial articles, or review studies. Animal studies and studies unrelated to *H. pylori*, dental caries, periodontitis, peri-implantitis, saliva, biofilms, or gastric infections were also excluded.

### 2.6. Data Extraction and Selection

Data extraction and selection were carried out systematically to ensure the quality and reliability of the information collected. In the first phase, the titles and abstracts of the studies identified in the databases were reviewed, excluding those that did not meet the inclusion criteria. In accordance with the PRISMA guidelines [16], studies that did not meet the predefined inclusion criteria were excluded through a structured selection process. The entire study selection procedure was carried out following the PRISMA standards and was visually summarized in a PRISMA flowchart. Two review authors independently screened the titles and abstracts to assess their relevance against the inclusion and exclusion criteria. The full texts of potentially eligible studies were retrieved and independently assessed by the same reviewers. Discrepancies regarding study eligibility were resolved by discussion, and, when a consensus was not reached, a third review author was consulted to make the final decision.

The full texts of the remaining studies were then assessed to confirm their eligibility according to the PECO criteria and the predefined inclusion and exclusion criteria. A standardized data extraction form was designed, which included general information such as the study (author, year, design) and population characteristics (sample size, age, sex), methods of detection of *H. pylori* (PCR, culture, antigen test), and oral diseases evaluated (dental caries, periodontitis, halitosis, etc.). The data were organized in a spreadsheet for ease of analysis and checked for accuracy against the full texts. To assess the methodological quality of the included studies, we used the NOS tool, which analyzes participant selection, group comparability, and outcome assessment. The extracted data were synthesized in summary tables, presenting the main characteristics of each study, the methods of detection of *H. pylori*, the oral diseases evaluated, and the key outcomes. This process ensured that the information collected was accurate and relevant, allowing for a rigorous synthesis of the available evidence to answer the research question.

### 2.7. Quality Analysis

The selected articles, presented in Table 1, were subjected to the NOS methodological evaluation scale to assess their methodological quality in three main domains: participant selection, group comparability, and outcome evaluation. In the screening domain, we assessed whether the studies included a representative sample of the target population, whether the controls were adequate, and whether the exposure (presence of *H. pylori* in the oral cavity) was reliably determined. For example, in studies such as Abdul et al. [17] and Chen et al. [18], it was verified that the inclusion and exclusion criteria were well defined and that the *H. pylori* detection methods (such as PCR or culture) were accurate and standardized. This ensured that the selection of participants was rigorous and minimized bias.

In the comparability domain, we looked at whether studies controlled for confounding factors such as age, sex, oral health status, or the presence of gastric infection. Studies such as Bouziane et al. [19] and Ozturk [20] demonstrated high comparability by adjusting key variables and using well-defined control groups. Finally, in the outcome assessment, we examined whether studies used valid methods to measure outcomes (oral diseases) and whether follow-up was sufficient. Thus, for example, the Urrutia-Baca et al. study [21] highlighted the use of quantitative techniques such as qPCR and a prospective design that allowed an evaluation of the effectiveness of interventions over time. The application of the NOS scale allowed most studies to be classified as “high quality”, reinforcing the validity of the findings reported in this systematic review.

**Table 1 microorganisms-13-01955-t001:** Methodological evaluation of the studies selected according to the Newcastle–Ottawa scale (NOS).

Author	Selection (Max. 4)	Comparability (Max. 2)	Result (Max. 3)	Total Score (Max. 9)	Methodological Quality
Abdul et al. [17]	4	1	2	7	High
Agarwal et al. [22]	4	1	2	7	High
Bouziane et al. [19]	4	1	2	7	High
Chen et al. [18]	4	1	2	7	High
Chen et al. [23]	4	2	3	9	High
El Batawi et al. [24]	3	1	3	7	High
Eskandari et al. [25]	4	1	2	7	High
Li et al. [26]	4	2	3	9	High
Medina et al. [27]	4	1	3	8	High
Mehdipour et al. [28]	4	2	3	9	High
Navabi et al. [29]	4	1	2	7	High
Ozturk [20]	4	2	3	9	High
Pataro et al. [30]	3	1	3	7	High
Ren et al. [31]	4	2	3	9	High
Sghaireen et al. [32]	3	1	2	6	Moderate
Sruthi et al. [33]	3	1	3	7	High
Tsimpiris et al. [34]	4	1	3	8	High
Tsimpiris et al. [35]	4	2	3	9	High
Urrutia-Baca et al. [21]	4	2	3	9	High
Zaric et al. [36]	3	1	2	6	Moderate
Zhang et al. [37]	4	2	3	9	High
Zheng et al. [38]	4	2	3	9	High

## 3. Results

### 3.1. Selection Process for Articles of Interest

The flowchart is shown in Figure 1. The search was conducted in January 2025 on four databases (PubMed, Scopus, Embase, and Cochrane), using a systematic strategy based on predefined inclusion and exclusion criteria. In PubMed, the following filters were applied: text availability (full text available), species (humans), date of publication (2010–2024), and MeSH terms such as “*Helicobacter pylori*”, “oral cavity”, “dental caries”, and “periodontitis”. We obtained 40 records. In Scopus, keywords such as “*H. oral pylori*”, “periodontal disease AND *H. pylori*”, and “biofilm AND *H. pylori*”, together with open-access filters, resulted in 94 records. Similarly adapted terms were used in Embase, with filters by date (2010–2024) and study design (observational, case–control, cross-sectional), identifying 79 additional records. The Cochrane Central search used the same terms, finding 85 records.

Initially, 298 records were identified, of which 110 were deleted due to duplication, leaving 188. After the review of titles and abstracts, 100 studies were excluded because they did not address the association between *H. pylori* and specific oral diseases. We assessed 88 full-text studies and excluded 66 for the following reasons: studies focusing on unrelated systemic pathologies (n = 35), no mention of *H. pylori* in the oral cavity (n = 20), and an unclear methodology or lack of quantitative data (n = 11). Finally, 22 studies that met all selection criteria and exhibited adequate methodological quality as evaluated by the NOS were included.

This process ensured rigorous and reproducible selection, prioritizing studies that evaluated the presence of *H. pylori* in dental plaque, saliva, or oral tissue and its association with caries, periodontitis, halitosis, or alterations of the oral microbiome in patients with or without gastritis. Heterogeneity in diagnostic methods (PCR, culture, antigen tests) and geographic variations in populations were recognized as key limitations, precluding a quantitative meta-analysis.

### 3.2. Study Characteristics

The study characteristics are summarized in Table 2. The studies reviewed cover a wide variety of methodological designs and demographic characteristics. In terms of the type of study, systematic reviews and meta-analyses predominate [17,18,19,20,29,31,35], followed by clinical studies, such as cases and controls [22,27,28,36], cross-sectional studies [3,4,5,6,7,8,9,10,11,12,13,14,15,16,17,18,19,20,21,22,24,27,28,29,31,35,36], and randomized clinical trials [21,23]. In vitro experimental research is also included [26,37]. In relation to the country of origin, China stands out with five studies [18,23,26,31,38], followed by Iran with three [25,28,29] and Saudi Arabia, India, and Greece with two each. Other countries represented are Argentina, Brazil, the United Arab Emirates, Morocco, Mexico, Serbia, and Turkey, with one study each.

In terms of sample size, studies vary significantly, from small groups such as 20 children [33] to large cohorts such as 54,036 participants in Zheng et al.’s study [38]. Some studies do not specify the sample size [26,37], while others include multiple studies with different sizes [17]. The average age of the participants also shows great diversity. For example, Mehdipoor et al.’s study [28] focused on children with an average age of 7.97 years, while others included adults with average ages between 42 and 55 [25,34]. Some studies do not specify the age [18,19,20].

Regarding samples in the oral cavity, most studies used dental plaque [22,25,28,32], saliva [27,30,34,36], or a combination of both. Other studies looked at specific samples such as carious lesions [24,33,38] or periodontal ligaments [26]. Regarding the presence of gastric infection, some studies exclusively included patients with infections [21,23,36], while others compared groups with and without infections [22,25,27]. Some focused on non-gastric niches [24,33]. Finally, the most common detection methods for *H. pylori* were PCR [17,22,24,30,36], labeled urea breath tests [19,20,23,38], and techniques such as RUT, ELISA, and culture [18,29,37]. Some studies used advanced methods such as electron microscopy [26] or stool antigen tests [32].

### 3.3. Summary of Results

-Relationship between oral *H. pylori* and gastritis

The studies reviewed show a clear trend towards the greater detection of *H. pylori* in the oral cavities of patients with gastritis compared to those without gastric infection. Studies [27,36] show high concordance between the presence of the organism in saliva or dental plaque samples and gastric infection, as detected by molecular methods. However, this association is not absolute, since [25] revealed that a significant percentage of patients with gastritis did not have *H. pylori* in the oral cavity, while some controls without gastritis harbored *H. Pylori*. Of particular interest are the findings in pediatric populations [24,33], where the pathogen was identified in deep caries lesions even in the absence of gastric disease, suggesting that the oral cavity may act as an independent reservoir in certain population groups.

-Distribution of *H. pylori* in different oral pathologies

When analyzing the different oral locations, dental plaque and saliva emerge as the main sites of detection of the microorganism. Studies like Sghaireen et al. [32] report particularly high prevalences in dental plaque [77.5%], while others, such as Tsimpiris et al. [34], find significantly lower figures [18%], possibly due to differences in the populations studied or in sampling techniques. In the context of periodontitis, the results are contradictory: while some studies suggest an association between the subgingival presence of *H. pylori* and periodontal disease [29], others fail to establish this link. On the other hand, research focused on dental caries, especially in the pediatric population, yields consistent results on the pathogen’s ability to colonize carious dentin, regardless of the patient’s gastric status, raising questions about its possible role in oral microbial ecology.

-Influence of detection techniques on findings

The marked heterogeneity in the diagnostic methods used represents a significant limitation when comparing the results between studies. PCR, although widely used, presents important variations in sensitivity depending on the specific technique used (nested PCR vs. qPCR), which could explain the discrepancies between the studies [34,36]. Urea breath tests, while useful in diagnosing gastric infections, cannot differentiate between the presence of the organism in the stomach and in the oral cavity. Techniques such as culture, successfully employed by Zhang et al. [37], are less sensitive but provide valuable information about the viability of the bacteria detected. This methodological diversity underscores the need to standardize screening protocols to allow for more robust comparisons between future investigations.

## 4. Discussion

The oral cavity, as an extragastric reservoir of *H. pylori*, has been extensively studied for its possible association with oral diseases in patients with or without gastritis, although the results remain controversial. The importance of this link lies not only in the local impact of the bacterium on oral health, such as its association with halitosis or lingual papilloma, whose symptoms improve significantly after gastric eradication [36], but also in its potential role as a source of reinfection and transmission.

Several studies support that the detection of *H. pylori* in the oral cavity is associated with both gastric infection and certain oral pathologies. In the case of periodontal disease, a meta-analysis has estimated the odds ratio (OR) to be 2.3 [38], while studies focused on periodontitis report stronger associations, with an OR of 3.42 (95% CI: 2.71–4.31) [39]. Likewise, it has been reported that the prevalence of oral *H. pylori* reaches approximately 45% in patients with gastric colonization, compared to 24% in those without gastric infection [40]. For peri-implantitis, the available data are scarce and do not allow us to establish robust estimates.

Regarding dental caries, the evidence is heterogeneous: some population-based studies have identified strong associations, e.g., an OR of 5 in a Japanese cohort [41]. In the case of saliva, an OR = 3.61 (95% CI 1.91–6.82) was found, which represents a strong association [40]. The association between *H. pylori* and supra- and subgingival biofilms is usually located in an OR range of 3 to 5, which corresponds to a moderate to high strength of association [40]. However, the overall certainty of the evidence remains limited due to methodological heterogeneity (differences in screening techniques, populations studied, and clinical definitions) and the predominance of observational studies.

The clinical implications of this oral reservoir are particularly relevant for therapeutic management. On the one hand, interventions such as periodontal therapy could increase the effectiveness of gastric eradication and reduce recurrences [20,31], although more studies are still needed to confirm this benefit [31]. On the other hand, the persistence of *H. pylori* in dental plaque, protected from systemic antibiotics [32], could explain treatment failures and reinfections [31,37]. In fact, combined strategies including oral hygiene with neutral electrolyzed water have been shown to be more effective than standard therapy alone, significantly reducing recurrence rates [21].

The connection between *H. pylori* and periodontal disease (PD) adds another layer of complexity. While some studies report a positive association, with an odds ratio (OR) of 3.50 in patients with gastric disease [18,35], others do not find a significant correlation [23,34]. This divergence could reflect differences in individual microbial profiles or diagnostic criteria. In addition, the high concordance of 82% between the presence of *H. pylori* in dental plaque and gastric mucosa [29] reinforces the idea that oral biofilm control is crucial, not only for oral health but also in preventing systemic spread [18,26].

It is also important to mention that recent microbiome studies and histopathological analyses show that the abundance and detectability of *H. pylori* tend to decrease in advanced mucosae or gastric cancer compared to active gastritis. The transition from chronic gastritis to atrophic gastritis and intestinalization generates a less favorable environment (loss of glands, reduced acid secretion, remodeling of the niche) for the survival of *H. pylori*, which favors substitution by other microbial communities associated with the tumor [39]. For example, a microbial profiling analysis in gastric tissue has shown a significant reduction in the abundance of *H. Pylori* in carcinoma samples compared to gastritis/precancer samples [40]. Therefore, although *H. pylori* is the primary cause of gastritis and a major indirect carcinogen, its prevalence and bacterial detectability may be higher in the gastritis/precancerous lesion phase, as in established cancer, which explains why studies comparing direct detections may find fewer *H. pylori* instances in tumor tissues than in inflamed gastric mucosae.

With respect to oral cancer, the current evidence is insufficient to consider *H. pylori* as a causal factor of oral cancer. It is reasonable to suggest that, although *H. pylori* can be found in the oral cavity and in some cases of carcinoma, the data are heterogeneous and are probably influenced by detection biases and confounding factors. Well-designed prospective studies (oral/gastric paired sampling, tests differentiating viable bacteria from free DNA, tight confounder control, and longitudinal analysis) are required before an etiological role can be attributed to *H. pylori* in oral carcinogenesis [41].

Finally, regarding therapeutics, if treatment failure can contribute to gastric reinfection, it is reasonable to consider stomatological interventions as part of a comprehensive approach. For example, oral hygiene measures and non-surgical periodontal treatments (such as scaling, root planing, and mouthwashes) have been linked to a higher eradication rate and a lower risk of recurrence of *H. pylori* in recent studies. In a recent systematic review, patients who received periodontal treatment in addition to standard antibiotic therapy showed significant improvements in the oral and gastric eradication of this bacterium [42]. In addition, adequate levels of oral hygiene (OHI ≤ 1.25) were associated with higher odds of success in the eradication of gastric (OR = 3.19) and oral (OR = 4.57) disease [43].

However, challenges such as the difficulty in eradicating *H. pylori* from dental plaque with conventional therapies [22,24,33] underscore the need for comprehensive approaches. Current evidence suggests that ignoring this oral reservoir could compromise treatment success, especially in populations with a high prevalence of coinfection, reaching 40% according to [32]. Future research studies should standardize diagnostic methods and explore dual strategies that combine gastric eradication with targeted oral interventions, such as mechanical and chemical plaque control [19,25].

Therefore, we suggest that, in patients with gastric *H. pylori* infection or persistent therapeutic failure, dentists should consider screening by PCR or antigen tests in saliva, plaque, or periodontal pockets and incorporate oral eradication strategies (intensive oral hygiene, antimicrobial mouthwashes, periodontal treatment) to complement medical treatment, as part of a multidisciplinary approach.

This systematic review has some limitations that should be considered when interpreting the results. The included studies show considerable methodological heterogeneity, both in the techniques used for the detection of *H. pylori* (PCR, culture, rapid urease test, histology, immunohistochemistry) and in the sites and protocols of oral sampling, which makes a direct comparison of the findings difficult. Cross-sectional observational designs predominate, with small sample sizes and without prior calculation of statistical power, which limits the ability to establish causal relationships. Likewise, geographical, socioeconomic, and cultural variability among the populations studied could influence the reported prevalence and oral hygiene practices.

## 5. Conclusions

Current evidence suggests that the oral cavity may be a reservoir of *H. pylori*, as well as the fact that there is a moderate strength of association between this pathogen and oral diseases such as caries and periodontitis and its presence in saliva and biofilms. In addition, oral hygiene and periodontal therapy can play an important role in eradicating and preventing the recurrence of gastric infections. However, more studies are needed to confirm these findings and determine the most effective strategies for the treatment of *H. pylori* in the oral cavity.

## Figures and Tables

**Figure 1 microorganisms-13-01955-f001:**
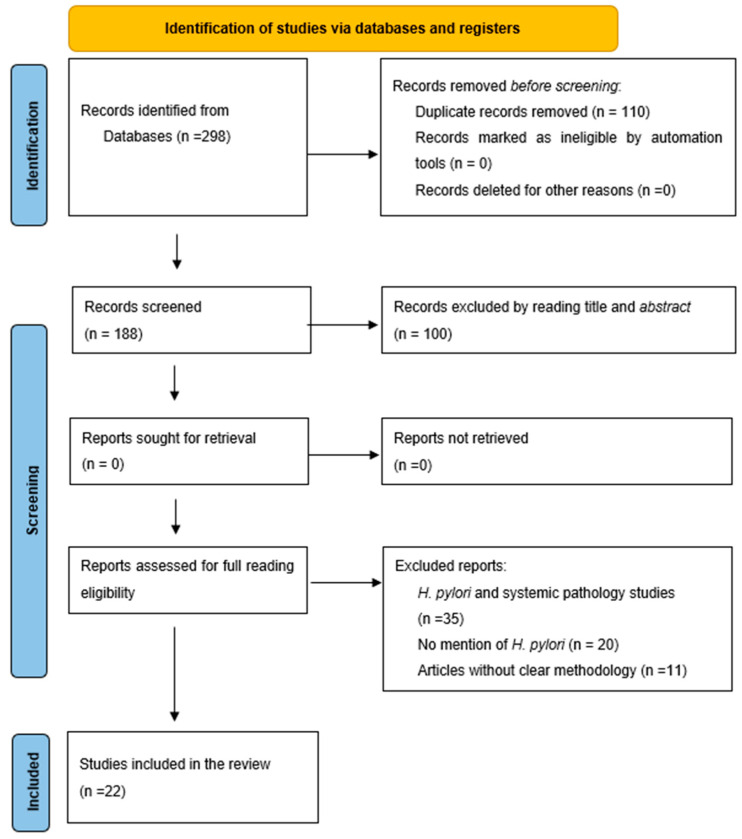
PRISMA diagram for study selection.

**Table 2 microorganisms-13-01955-t002:** General characteristics of the articles included in the systematic review.

N°	Author/Year	Type of Study	Country	Sample Size	Average Age	Samples in the Oral Cavity	With Gastric Infection	H. pylori Detection Method	Main Results
1	Abdul et al., 2023 [17]	Systematic review	Saudi Arabia	15 included studies (ranging in size from 10 to 1050 subjects)	Variable according to study (between 3 and 74 years)	Miscellaneous (dental plaque, saliva, oral specimens)	Includes studies with and without gastric infection	Miscellaneous: mainly PCR, plus endoscopy, antigen, and checkerboard DNA-DNA	Associated oral *H. pylori* with increased caries, even without gastric infection.
2	Agarwal et al., 2012 [22]	Clinical study (cases and controls)	India	50 patients (30 cases and 20 controls)	Cases: 46.2 ± 11.44 years; controls: 44.5 ± 11.36 years	Subgingival plaque	Cases: with infection. Controls: no symptoms.	PCR (16S rRNA) and culture	*H. pylori* was found in the subgingival plaque, related to periodontitis and dyspepsia.
3	Bouziane et al., 2012 [19]	Meta-analysis and systematic review	Morocco	298 patients	Unspecified	Unspecified	Yes	Urea 13C breath test and PCR	Periodontal therapy reduced recurrence of gastric *H. pylori* by 63%.
4	Chen et al., 2019 [18]	Meta-analysis	China	6200 patients	Unspecified	Unspecified	It includes both patients with gastric disease and the general population.	RUT, PCR, and ELISA (according to study)	Oral *H. pylori* increases the risk of periodontal disease (OR 2.31), especially in gastric patients (OR 3.50).
5	Chen et al., 2023 [23]	Randomized clinical trial	China	160	Unspecified	Not applicable	Yes	Urea 13C breath test	Periodontal therapy improves *H. pylori* eradication (87% vs. 75%) and reduces recurrence.
6	El Batawi et al., 2020 [24]	Cross-sectional study	United Arab Emirates	48 children	Approximately 5.2 years (range 4–7)	Dentin samples from cavitated carious lesions	No (a non-gastric niche is studied)	*PCR*	*H. pylori* in childhood caries (30%) suggests oral reservoir.
7	Eskandari et al., 2010 [25]	Observational research study	Iran	67 patients	42.3 ± 12.52 years	Biofilm (supra- and subgingival)	23 patients with gastritis; 44 without gastritis	*PCR*	*H. pylori* in dental plaque (5.9%) associated with gastritis (*p* = 0.012), a possible source of reinfection.
8	Li et al., 2019 [26]	In vitro experimental study	China	Not applicable (hPDLF culture)	Range 14–25 years	Periodontal ligament	Not applicable	Observation of invasion by transmission electron microscopy (TEM)	*H. pylori* inhibits the proliferation of periodontal fibroblasts via Cdc25C/CDK1/cyclin B1, causing G2 arrest.
9	Medina et al., 2010 [27]	Case–control study	Argentina	98 patients (43 cases and 55 controls)	43.7 years	Saliva and dental plaque	Yes (cases with digestive symptoms; 43 patients with positive biopsy)	PCR in oral specimens; histology (Giemsa and H&E) in gastric biopsies	*H. pylori* in the oral (18.4%) and gastric (88.4%) cavities correlates with periodontal disease (*p* < 0.05), risk of reinfection.
10	Mehdipour et al., 2022 [28]	Case–control study	Iran	72 (36 cases and 36 controls)	7.97 ± 1.83 years	Dental plaque, lower molars	No	PCR (16S rRNA gene detection and virulence analysis: *vacAm1*, *vacAs1*, *dupA*)	*H. pylori* in dental plaque in children (20.8%), not related to caries.
11	Navabi et al., [29]	Meta-analysis and systematic review	Iran	1861 patients (745 men and 790 women)	42.8 ± 7.4 years	Dental plaque	Coinfection in plaque and stomach: 49.7%	Several (PCR, RUT, culture, serology, histology) depending on each study.	50% with oral–gastric coinfection, but doubtful role in reinfection.
12	Ozturk, 2021 [20]	Meta-analysis	Türkiye	1450 participants	Unspecified	Not assessed (meta-analysis study)	Yes (patients undergoing eradication therapy)	Breath test with 13C urea and other methods reported in clinical trials	Periodontal treatment improves gastric eradication (OR = 4.11) and reduces recurrence (OR = 5.36).
13	Pataro et al., 2016 [30]	Cross-sectional study	Brazil	154 participants	Adults (18–65 years old)	Saliva and tongue scraping	*H. pylori* detected in 83.3% of gastric biopsies	PCR	Obese people have a high presence of periodontal pathogens and *H. pylori* in the mouth/stomach. Bariatric surgery reduces bacteria in the stomach but increases them in the mouth.
14	Ren et al., 2016 [31]	Systematic review and meta-analysis	China	691 participants	17–78 years	Saliva, dental plaque, periodontal pockets	Yes	Urea breath test (13C or 14C), RUT, histology, PCR	Periodontal therapy improves gastric *H. pylori* eradication and reduces recurrence, but more studies are needed.
15	Sghaireen et al., 2017 [32]	Cross-sectional study	Saudi Arabia	120 male students	22.37 ± 1.50 years	Dental plaque (PCR)	Yes (93 students, 77.5%)	Stool antigen test *(Helicobacter* Antigen Quick, GA Inc., Dresden, Germany)	Significant association between *H. pylori*, cotinine (tobacco), and dental caries.
16	Sruthi et al., 2023 [33]	Cross-sectional study	India	20 children	4.85 years	Deep carious lesions (dentin)	No (only children without gastric diseases)	RT-PCR (reverse transcription and polymerase chain reaction)	*H. pylori* in 70% of children with severe caries, related to an increase in FMDD.
17	Tsimpiris et al., 2022 [34]	Clinical Study	Greece	65 (33 with periodontitis and 32 controls)	55.5 years (average)	Saliva (rtPCR)	6 patients with periodontitis and 7 controls	Saliva: rtPCR (real-time PCR)	No association was found between chronic periodontitis and oral/gastric *H. pylori.*
18	Tsimpiris et al., 2023 [35]	Systematic review and meta-analysis	Greece	818 (total subjects in meta-analysis)	Unspecified	Subgingival plaque	Not specified (varies between included studies)	RUT (rapid urease test) and PCR (polymerase chain reaction)	There is no *H. pylori* in saliva, but periodontal improvement after gastric eradication.
19	Urrutia-Baca et al., 2024 [21]	Randomized clinical trial	Mexico	100 participants	43.34 years (average)	Dental plaque (qPCR)	100 patients (all with gastric infection)	Dental plaque: qPCR (quantitative PCR)	New systemic therapy achieved gastric eradication in 84–96% and fewer recurrences.
20	Zaric et al., 2015 [36]	Clinical intervention study (double-blind, case–control)	Serbia	98 (66 with gastric infection)	Range: 19–78 years old	Saliva (nested PCR)	66 patients (with gastric infection)	Saliva: nested PCR	Gastric eradication significantly reduced halitosis and the lining of the tongue.
21	Zhang et al., 2018 [37]	In vitro experimental study	China	Unspecified	Unspecified	Dental plaque, saliva	Yes (*H. pylori*-positive)	Culture and PCR	*H. pylori* alters the oral balance, favoring *Streptococcus mutans* and promoting caries.
22	Zheng et al., 2014 [38]	Cross-sectional study (population analysis)	China	54,036	46.3 years	Dental plaque, saliva	Yes (46.97% positive)	^13^C-labeled urea breath test	*H. pylori* is associated with dental calculus and tooth mobility, as well as factors such as smoking and obesity.

## Data Availability

No new data were created or analyzed in this study. Data sharing is not applicable to this article.

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
