# Peer review of "Association of Helicobacter pylori as an Extragastric Reservoir in the Oral Cavity with Oral Diseases in Patients with and Without Gastritis—A Systematic Review"

_microorganisms, 2025, doi:10.3390/microorganisms13081955_

Round 1
Reviewer 1 Report
Comments and Suggestions for Authors
A timely manuscript on a debatable issue that the oral cavity serves as a repository for reinfections of H. pylori. While the manuscript is interesting, several issues were found:
- Unclear what the goal of this manuscript is.
- Table 2 under the "Author/Year" is that the number in the reference section?
- Lack of medical information, such as GERD, could explain the presence of H. pylori in the oral cavity.
- Plenty of punctuation errors throughout the manuscript, especially when using brackets for citation numbers.
- Missing in the discussion/conclusion is a limitation of the study.
Reviewer 2 Report
Comments and Suggestions for Authors
The manuscript presents the relation between Helicobacter pylori infection and diseases of the oral cavity. Also entitled as a systematic review, it is rather a narrative one. The period of literary research (2010-2025) is too short, significanr articles have been published also before 2010.
The number of articles screened seems to be rather short, there are 564 papers on Pubmed with the search words +Helicobacter pylori and oral cavity". The relationship between H pylori and oral cancer is not addressed. Also the authors claim that the material is presented according to the PRISMA statement, this is not the case. Another major drawback is that there are no numerical data presented (percentages, odds ratios, relative risk) and the quality of evidence of some associations is not mentioned. No diagnostic and therapeutic advices are presented for treatment of oral H pylori infection (regular stomatologic assessment of those infected? eradication? mouth disinfection? screening for H pylori by stomatologists?other?) .
Comments on the Quality of English Languagethe reviewer is not a native English speaker
Round 2
Reviewer 1 Report
Comments and Suggestions for Authors
My concerns were addressed by the authors. It is a manuscript with a limited number of studies, which may be because this area remains controversial in the biology of H. pylori.
The Editorial Office can address other issues associated with typos and style in the references.